# Rifampicin and isoniazid drug resistance among patients diagnosed with pulmonary tuberculosis in southwestern Uganda

**Lisa Nkatha Micheni**[1,2], **Kennedy Kassaza**[1], **Hellen Kinyi**[3], **Ibrahim Ntulume**[2], **Joel Bazira**[1]*

**1** Department of Microbiology, Faculty of Medicine, Mbarara University of Science and Technology, Mbarara, Uganda, **2** Department of Microbiology and Immunology, Faculty of Biomedical Sciences, Kampala International University Western Campus, Bushenyi, Uganda, **3** Department of Biochemistry, School of Medicine, Kabale University, Kabale, Uganda

* jbazira@must.ac.ug

## Abstract

Multidrug-resistant tuberculosis (MDR-TB) has become a major threat to the control of tuberculosis globally. Uganda is among the countries with a relatively high prevalence of tuberculosis despite significant control efforts. In this study, the drug resistance of *Mycobacterium tuberculosis* to rifampicin (RIF) and isoniazid (INH) was investigated among patients diagnosed with pulmonary tuberculosis in Southwestern Uganda. A total of 283 sputum samples (266 from newly diagnosed and 17 from previously treated patients), collected between May 2018 and April 2019 at four different TB diagnostic centres, were assessed for RIF and INH resistance using high-resolution melt curve analysis. The overall prevalence of monoresistance to INH and RIF was 8.5% and 11% respectively, while the prevalence of MDR-TB was 6.7%. Bivariate analysis showed that patients aged 25 to 44 years were at a higher risk of developing MDR-TB (cOR 0.253). Furthermore, among the newly diagnosed patients, the prevalence of monoresistance to INH, RIF and MDR-TB was 8.6%, 10.2% and 6.4% respectively; while among the previously treated cases, these prevalence rates were 5.9%, 23.5% and 11.8%. These rates are higher than those reported previously indicating a rise in MTB drug resistance and may call for measures used to prevent a further rise in drug resistance. There is also a need to conduct frequent drug resistance surveys, to monitor and curtail the development and spread of drug-resistant TB.

## Introduction

Tuberculosis (TB) is a severe infectious disease associated with high rates of mortality and morbidity worldwide despite intense efforts. One of the major factors sustaining the TB epidemic is the increasing number of *Mycobacterium tuberculosis* (MTB) strains that do not respond to TB therapy [1–4]. Currently, the standard TB treatment regimen combines four first-line antibiotics, isoniazid (INH), rifampin (RIF), pyrazinamide (PZA), and ethambutol (EMB), which renders a patient noncontagious when they are properly administered.

**Funding:** The authors received no specific funding for this work.

**Competing interests:** The authors have declared that no competing interests exist.

However, inadequate treatment can result in drug-resistant MTB among these patients (acquired resistance), and those resistant strains can be transmitted to other individuals (primary resistance). MDR-TB, described as resistance to both INH and RIF (the two most effective antibiotics for TB treatment) is of great concern because second-line antimycobacterial drugs require long-term administration, are expensive, and have a wide variety of side effects [5]. MDR-TB emerged in the early 1990s and continues to rise each year [1, 6]. In 2017 the world health organization (WHO) reported that approximately 4.1% of new TB cases and 19% of previously treated cases had developed multidrug resistance [7]. In Uganda, the prevalence of MDR-TB is estimated at 12% and 1% among previously-treated and newly diagnosed patients respectively by 2019 [8]. Continuous screening of antibiotic resistance in TB patients, focusing primarily on INH and RIF, is critical since these two medications are the backbone of TB treatment [9]. Resistance to RIF is related to mutations in the rpoB gene, which encodes the β-subunit of bacterial DNA-dependent RNA polymerase, while INH resistance is a result of two pathways involving large mutations in the katG (encodes a catalase-peroxidase enzyme which converts INH to its active form) and inhA genes [10]. An amino acid substitution on codon 315 of katG due to mutation accounts for 42%–95% of INH resistance [11] while a mutation at the promoter region of the inhA gene encoding the enoyl-ACP- reductase, confers 6%–34% of INH resistance in MTB strains [11]. There are several techniques applied in the detection of drug resistance. The conventional drug susceptibility testing (DST) systems which focus on the culture of MTB is time-consuming as it takes approximately 2 to 4 weeks due to the slow growth of this bacteria [12]. Other approaches such as microscopic observation drug susceptibility (MODS; Hardy Diagnostics) and thin-layer agar (TLA; NanoLogix, Inc.) assays are quicker but are highly operator dependent, expensive and technically challenging in resource-limited areas [12]. The establishment of Molecular techniques that identify mutations associated with drug resistance drastically decreases the diagnostic delay and, in some cases, may prove to be more specific than phenotypic DST [13]. High-resolution melt curve analysis (HRMA) is a molecular technique that can be used to detect subtle genetic mutations conferring drug resistance in MTB and is based on the detection of a variation of the deoxyribonucleic acid (DNA) sequence demonstrated by fluorescence changes in the melting temperature of a double-stranded DNA amplicon after real-time polymerase chain reaction (qPCR) [14]. This overall sensitivity and specificity of HRMA is approximately 94% and 99%, respectively in the detection of MDR-TB [15–17]. It is also simple, quick and easy to perform hence can be used to screen a vast number of samples within a limited time. This study utilized the HRMA technique to screen for RIF and INH among patients diagnosed with pulmonary tuberculosis (PTB) in Southwestern Uganda, a region heavily affected by the TB/HIV epidemic [18, 19].

## Materials and methods

### Ethics approval and consent to participate

Ethical clearance for this study was obtained from the Mbarara University of Science and Technology Institutional Review Board committee (ref. 13/08-17) and clearance was acquired from the Uganda National Council for Science and Technology Research under (ref. HS2379). Permission was also obtained from the office of the Prime minister to access the refugee camps while the health facility administrators granted permission to access their facilities. Participants consented to enroll in the study upon completing an informed consent form.

### Study setting and population

This was a cross-sectional study conducted in the Southwestern region of Uganda between May 2018 and April 2019. The Cochran sampling technique [20] was used to determine the

sample size and a total of 283 consenting patients were recruited. Four recruitment centres were utilized; two regional referral hospitals (Kabale and Mbarara regional referral hospitals) and two health centres located within two refugee camps (Oruchinga and Nakivale health centre IV). Patients aged ≥ 18 years, diagnosed with PTB for the first time and those previously treated for TB at any of the four study locations were eligible to participate in this study. Non-sputum samples from patients with extrapulmonary TB were excluded.

## Data and sample collection

Sputum samples from eligible patients were collected and confirmed for TB using either smear microscopy or Cepheid GeneXpert. Besides the sample collection, demographic characteristics such as age, gender, ethnicity, place of residence, level of income, HIV status before enrollment were obtained from the eligible patients using a standard clinical form. Information about certain risk factors for exposure to resistant strains including imprisonment, previous history of TB treatment and history of recent migration into the country were also collected. The sputum samples were refrigerated at 4˚C at the recruitment centres, for not more than 72 hours, and later transported in a cold box to Mbarara University of Science and Technology Genomics and Translational Laboratory for processing and molecular analysis.

## DNA extraction and confirmation of MTB in sputum samples

The genomic DNA from each patients' sputum sample was processed by standardized protocols [21, 22]. The samples were then screened and confirmed as MTB by detection of a 123-bp fragment of the *IS*6110 gene which is common among the members of the MTBC.

## Real-time PCR and high-resolution melting analysis

Three pairs of primers (S1 Table) targeting specific sites at which mutations associated with RIF and INH drug resistance were utilized. One of each of the primer pairs, amplifying either the rpoB [the 81-bp RIF resistance determining region (RRDR)], katG [the -315 site (INH resistance)] and the promoter region of InhA [the -8 and -15 sites (INH resistance)]. The amplification was carried out in a B*io-Rad CFX96* Touch™ using a Lunar® Universal genotyping master mix. All qPCR assays were performed in a final volume of *20 μl reaction* mixture containing the following components per reaction: 1.25 μl (0.5 mM final concentration) of each primer, 12.5 μl of 2x HRM PCR master mix, 2 μl of PCR water and 3 μl (5–50 ng) of genomic DNA. The B*io-Rad CFX96* Touch™ *Real-Time* PCR Detection *System* was programmed for PCR amplification and a melting curve stage. The thermal cycling parameters were 10 min at 95˚C of pre-PCR stage and an amplification stage of 40 cycles consisting of 95˚C for 5 secs and 10 secs at 60˚C. The qPCR amplification was followed by melt curve analysis, which was initiated by a holding step at 65˚C for 5 sec (to allow reassociation of DNA), followed by a slow temperature increase to 95˚C at a rate of 0.1˚C/s with continuous fluorescence data acquisition. The wild-type MTB (H37Rv), and nuclease-free water were included in each run as positive control and negative controls respectively. The HRMA curve was analysed using B*io-Rad CFX96* Touch™ manager software. Converting the wild-type melting profile to a horizontal line and normalizing the melting profiles of the analyzed samples against the wild-type profile provided the difference temperature plots (S1 Fig).

## Statistical analysis

Data were entered into Microsoft Excel 2010 software and then exported to SPSS version 25 (IBM, Chicago, USA) for analysis. The Chi-square test was computed to determine significance for observed differences. The threshold for statistical significance was set at $p \leq 0.05$.

## Results

A total of 283 sputum samples were analyzed, out of which 266 were from newly diagnosed patients and 17 previously treated patients. The median age of all patients was 36 years and the majority were male TB patients (73.1%). 5.7% and 6.4% of the samples were collected from prisoners and refugees respectively (Table 1).

The overall monoresistance to rifampicin and isoniazid was found in 11% (95% CI: 0.077–0.150; p, 0.087) and 8.5% (95% CI: 0.056–0.123; p, 0.692) of the patients, respectively. Monoresistance to rifampicin and isoniazid was found in 11% (95% CI: 0.077–0.150; p, 0.087) and 8.5% (95% CI: 0.056–0.123; p, 0.692) of all the patients, respectively. Resistance to RIF and INH among newly diagnosed patients was 10.2% and 8.6%, while among previously treated patients, resistance to RIF and INH was 23.5% and 5.9% respectively. Furthermore, 4.9% of the samples from newly diagnosed with INH monoresistance, were found to have mutations in the InhA region while 8.6% had mutations in the katG region, a condition that can lead to phenotypic isoniazid drug resistance [2]. The overall resistance to both drugs (MDR-TB) was 6.7% (95% CI: 0.026–0.075; p, 0.982) among all patients (6.4% among newly diagnosed and 11.8% among previously treated patients). Of the cases with MDR-TB, 6.4% had mutations in both the rpoB and KatG regions, while 3.9% had mutations in both the rpoB and InhA sites (Table 2).

Bivariate analysis revealed that the previously treated patients were more likely to have MDR than the newly diagnosed patients (cOR: 1.092; 95% CI: 0.137–8.737). This relation, however, was not statistically significant (p >0.05) given the very low sample size of the previously treated patients sample". Furthermore, there was a substantial correlation between age

**Table 1. Demographic characteristics of patients enrolled for the study between May 2018 and April 2019.**

| Characteristic | Category | N = 283; n (%) |
|---|---|---|
| Age | ≤ 24 | 45 (15.9) |
| | 25–44 | 158 (55.8) |
| | 45–64 | 57 (20.1) |
| | ≥ 65 | 23 (8.1) |
| Gender | Male | 207 (73.1) |
| | Female | 76 (26.9) |
| HIV Status | Positive | 76 (26.9) |
| | Negative | 78 (27.6) |
| | Unknown | 129 (45.6) |
| Level of income | High | 23 (8.1) |
| | Low | 260 (91.9) |
| Prisoner | No | 267(94.3) |
| | Yes | 16 (5.7) |
| Refugee | No | 246 (86.9) |
| | Yes | 37 (13.1) |
| TB in the past | No | 266 (94) |
| | Yes | 17 (6) |

**Table 2. Frequency of rpoB, KatG and InhA mutations in *Mycobacterium tuberculosis* causing PTB in southwestern Uganda; May 2018 and April 2019.**

| Frequency of mutation (pattern of resistance) | Newly diagnosed | Previously-treated | All cases | p-value* |
|---|---|---|---|---|
| | N = 266 | N = 17 | N = 283 | |
| **Monoresistance to** | **n (%)** | **n (%)** | **n (%); 95% CI** | |
| **RIF** | **27 (10.2)** | **4 (23.5)** | **31 (11.0); (0.077–0.150)** | **0.087** |
| **INH£** | **23 (8.6)** | **1 (5.9)** | **24 (8.5); (0.056–0.123)** | **0.692** |
| i. InhA mutations | 13 (4.9) | 0 (0.0) | 13 (4.6); (0.026–0.075) | 0.351 |
| ii. KatG mutations | 23 (8.6) | 1 (5.9) | 24 (8.5); (0.056–0.123) | 0.692 |
| **MDR¥** | **17 (6.4)** | **2 (11.8)** | **19 (6.7); (0.039–0.097)** | **0.982** |
| i. rpoB & KatG mutations | 17 (6.4) | 1 (5.9) | 18 (6.4); (0.039–0.097) | 0.934 |
| ii. rpoB & InhA mutations | 9 (3.4) | 2 (11.8) | 11 (3.9); (0.016–0.058) | 0.441 |

Mutation patterns detected using HRMA:

£One or more mutations leading to single drug (INH) resistance;

¥More than one mutation leading to resistance of both drugs (multidrug resistance);

*p-value obtained by chi-square statistic.

and drug resistance: patients aged 25 to 44 years old (cOR 0.253; 95% CI: 0.070–0.922; p = 0.037) were more likely to have MDR-TB. Among all the patients included in this study, 54.4% of them had their HIV status known at the time of the study, with 26.9% of them being HIV positive. There was no substantial correlation between MDR-TB HIV infection and this category of patients (p = 0.324). No other variables were observed to be associated with MDR-TB in the region (Table 3).

## Discussion

This study sought to determine the prevalence of RIF and INH resistance among PTB patients in Southwestern Uganda. DNA obtained from 283 sputum samples (each from an individual patient) were analysed. The overall prevalence of RIF and INH monoresistance was found to be at 11% and 8.5% respectively. This was significantly higher than that of a similar study carried out in part of this region between May 2007 and April 2008 which found RIF and INH monoresistance rates were at 4.8 and 3.2 per cent, respectively [23]. Studies have shown that an increase in drug resistance levels can be attributed to a variety of factors, including delayed diagnosis, non-continuous drug resistance surveillance [24, 25], and TB medication stock-outs in health facilities [19] resulting in treatment interruptions and the spread of drug-resistant strains in the population. Furthermore, studies have also shown that health care workers especially those involved in tuberculosis infection control, diagnosis, and treatment are a risk factor for resistance to any anti-TB medication [25]. However, studies are needed to investigate the role of such factors, and any other ones, in the rise of drug resistance in this region.

Our findings show that RIF monoresistance was higher (23.5%) in those patients with a history of prior TB treatment as compared to 10.2% of those newly diagnosed patients, while the rate of MDR was 6.4% and 11.8% in newly diagnosed and previously treated patients, respectively. Studies have shown that previous TB drug exposure, especially in inadequate or inappropriate doses, can result in an increased risk of developing mono-drug or multidrug-resistant tuberculosis [25–28]. The findings of this study of higher prevalence of MTB drug resistance among previously treated patients is consistent with findings of other Ugandan studies [25, 29, 30] and other neighbouring countries [4, 31, 32]. These high levels of drug resistance among previously treated patients raise concerns about TB treatment compliance and the successful application of drug resistance preventive measures such as directly observed

**Table 3. Bivariate analysis of factors associated with multidrug resistance among PTB patients in southwestern Uganda.**

| Risk factor | RIF/INH Susceptible, n (%) | RIF/INH resistant, n (%) | cOR (95% CI) | P-value |
|---|---|---|---|---|
| **Age (years)** | | 0.141 | | |
| ≤ 24 | 43 (95.6) | 2 (4.4) | 0.221 (0.037–1.312) | 0.097 |
| 25–44 | 150 (94.9) | 8 (5.1) | 0.253 (0.070–0.922) | 0.037* |
| 45–64 | 53 (93.0) | 4 (7.0) | 0.358 (0.081–1.577) | 0.175 |
| ≥ 65 | 19 (82.6) | 4 (17.4) | 1.000 | - |
| **Gender** | | 0.647 | | |
| Male | 193 (93.2) | 14 (6.8) | 1.306 (0.416–4.098) | 0.648 |
| Female | 72 (94.7) | 4(5.3) | 1.000 | |
| **Level of income** | | 0.632 | | |
| low | 244 (93.8) | 16 (6.2) | 0.689 (0.148–3.199) | 0.634 |
| High | 21 (91.3) | 2 (8.7) | 1.000 | |
| **HIV status** | | 0.539 | | |
| Positive | 72 (94.7) | 4 (5.3) | 0.968 (0.274–3.422) | 0.960 |
| Negative | 71 (91.0) | 7 (9.0) | 1.718 (0.579–5.099) | 0.329 |
| Unknown | 122(94.6) | 7 (5.4) | 1.000 | |
| **TB in the past** | | 0.934 | | |
| No | 249 (93.6) | 17 (6.4) | 1.092 (0.137–8.737) | 0.934 |
| Yes | 16 (94.1) | 2 (11.8) | 1.000 | |
| **Refugee** | | 0.328 | | |
| No | 229 (93.1) | 17 (6.9) | 2.672 (0.345–20.701) | 0.347 |
| Yes | 36 (97.3) | 1 (2.7) | | |
| **Prisoner** | | 0.091 | | |
| No | 249 (93.3) | 18 (6.7) | - | - |
| Yes | 16 (100.0) | 0 | - | - |

Variables included in the bivariate model: age, gender, HIV status, level of income, imprisonment, refugee status and previous history of TB;

*Statistically significant at 95% level of confidence;

cOR: adjusted odds ratio.

therapy (DOT) [9, 19]. More research, however, is needed to better understand the underlying causes of this greater occurrence of drug resistance among this category of patients in this region. There is also a need to reinforce and ensure strict compliance with the various policies developed by the Ministry of Health and the National Tuberculosis and Leprosy Program (NTLP).

Our study showed no significant association between patient characteristics such as imprisonment, refugee status, or HIV status and the development of MDR-TB, despite some studies showing that such factors are a strong independent risk factor for MDR-TB [4, 27, 33, 34]. For example, TB patients co-infected with HIV [35, 36] and the refugees [37, 38] are more likely to develop MDR-TB. Nevertheless, our results are comparable to those of a national survey of drug-resistant TB in Uganda [25] and a recent study in Arua [24], which found no significant association between HIV-TB co-infection and the development of MDR-TB. Nonetheless, integrated HIV/TB management is important for both diseases' management. Disaggregated by age, our study showed that the patients aged 25 to 44 years had higher levels of MDR-TB (cOR 0.253; 95% CI: 0.070–0.922), suggesting that this category of patients are at a higher risk of having MDR-TB than the older population.

## Study limitations

We did not test for all the possible mutations associated with rifampicin and isoniazid resistance. However, we targeted the most dominant mutations associated with rifampicin and isoniazid resistance, hence likely that we identified a majority of the resistance to these drugs in the region. Another significant limitation of our analysis is the small sample size of the patients who have previously been treated for tuberculosis, which limits the precision with which we can estimate the relationship between prior treatment experience and the likelihood of developing drug resistance. Furthermore, HIV testing was not conducted on patients who did not know their HIV status at the time of the study, which could not enable us to conclude on HIV as a risk factor for drug resistance.

## Conclusion

There is a relatively high prevalence of MDR-TB among PTB patients in Southwestern Uganda since settings with an MDR-TB prevalence of more than 3% especially among newly treated patients are considered as having a high MDR-TB burden (WHO 2015). This highlights the value of continuous drug resistance screening for all TB patients, as well as the strengthening of drug resistance control measures.

## Supporting information

**S1 Table. Primer pairs utilized in the high-resolution melting temperature curve analysis.** (TIF)

**S1 Fig. An illustration of the RT-PCR (B*io-Rad CFX96* Touch™) high-resolution melting temperature curve analysis using rpoB primers.** a) melt temperature of the target region b) derived melting curve. Green = sample; Black = negative control and Red = Positive control. (TIF)

## Acknowledgments

We gratefully acknowledge the technical help provided by the TB laboratory staff of Kabale and Mbarara regional referral hospitals, Nakivale health centre, and the Genomics and Translational laboratory of Mbarara University of Science and Technology.

## Author Contributions

**Conceptualization:** Lisa Nkatha Micheni, Joel Bazira.

**Data curation:** Lisa Nkatha Micheni, Kennedy Kassaza, Ibrahim Ntulume.

**Formal analysis:** Lisa Nkatha Micheni, Kennedy Kassaza, Hellen Kinyi, Ibrahim Ntulume.

**Funding acquisition:** Lisa Nkatha Micheni.

**Investigation:** Lisa Nkatha Micheni, Joel Bazira.

**Methodology:** Lisa Nkatha Micheni, Kennedy Kassaza, Joel Bazira.

**Project administration:** Joel Bazira.

**Resources:** Lisa Nkatha Micheni, Joel Bazira.

**Supervision:** Joel Bazira.

**Visualization:** Lisa Nkatha Micheni.

**Writing – original draft:** Lisa Nkatha Micheni.

**Writing – review & editing:** Lisa Nkatha Micheni, Kennedy Kassaza, Hellen Kinyi, Ibrahim Ntulume, Joel Bazira.

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
