## [Decision Letter · Decision Letter 0]

11 May 2021

PONE-D-21-11660

Rifampicin and Isoniazid drug resistance among patients diagnosed withpulmonary tuberculosis in southwestern Uganda

PLOS ONE

Dear Dr. Bazira,

Thank you for submitting your manuscript to PLOS ONE. After careful consideration, we feel that it has merit but does not fully meet PLOS ONE’s publication criteria as it currently stands. Therefore, we invite you to submit a fully major revised version of the manuscript that addresses the points raised during the review process.

The paper was shown to contains many flaws and must be entirely revised. Please note that we ask you to take all comments of the reviewers into consideration, since they are all experts in the field and raised the same concerns. A full re-review of the second version will be done by the same reviewers.

We look forward to receiving your revised manuscript.

Kind regards,

Christophe Sola, Pharm.D. Ph.D.

Academic Editor

PLOS ONE

Journal Requirements:

2. In your Methods section, please provide additional information about the participant recruitment method and the demographic details of your participants. Please ensure you have provided sufficient details to replicate the analyses such as:

- a description of any inclusion/exclusion criteria that were applied to participant recruitment

- a statement as to whether your sample can be considered representative of a larger population

- a description of how participants were recruited.

4. Please ensure you have thoroughly discussed any potential limitations of this study within the Discussion section, including the potential impact of confounding factors.

5.  We note that Figure S1 in your submission contain map images which may be copyrighted. All PLOS content is published under the Creative Commons Attribution License (CC BY 4.0), which means that the manuscript, images, and Supporting Information files will be freely available online, and any third party is permitted to access, download, copy, distribute, and use these materials in any way, even commercially, with proper attribution. For these reasons, we cannot publish previously copyrighted maps or satellite images created using proprietary data, such as Google software (Google Maps, Street View, and Earth). For more information, see our copyright guidelines: http://journals.plos.org/plosone/s/licenses-and-copyright.

5.1.    You may seek permission from the original copyright holder of Figure S1 to publish the content specifically under the CC BY 4.0 license. 

5.2.    If you are unable to obtain permission from the original copyright holder to publish these figures under the CC BY 4.0 license or if the copyright holder’s requirements are incompatible with the CC BY 4.0 license, please either i) remove the figure or ii) supply a replacement figure that complies with the CC BY 4.0 license. Please check copyright information on all replacement figures and update the figure caption with source information. If applicable, please specify in the figure caption text when a figure is similar but not identical to the original image and is therefore for illustrative purposes only.

Reviewers' comments:

Reviewer's Responses to Questions

**Comments to the Author**

1. Is the manuscript technically sound, and do the data support the conclusions?

Reviewer #1: Partly

Reviewer #2: Partly

2. Has the statistical analysis been performed appropriately and rigorously? 

Reviewer #1: No

Reviewer #2: No

3. Have the authors made all data underlying the findings in their manuscript fully available?

Reviewer #1: Yes

Reviewer #2: Yes

4. Is the manuscript presented in an intelligible fashion and written in standard English?

Reviewer #1: Yes

Reviewer #2: No

5. Review Comments to the Author

Reviewer #1: While the study appears sound, the data presentation is flawed. Specifically according to table 3, 258 of the susceptible patients have a history of past TB, while in the results it is stated the contrary (266 new and 17 relapses). As a result after statistical analysis, the authors erroneously claim in line 160-161,

that "Newly diagnosed patients were more likely to have MDR than patients who had previously received treatment". Also in table 3, the number of MDR cases taken for statistical analysis is 8 for all variables. The study found a total of 19 MDR cases. It is not clear what the number 8 corresponds to; so is unclear the whole statistical analysis based in table 3.

I advise the authors to improve table 2 by putting only numbers of samples with mutations found for each of the mutations, instead of all (mutations found and not found). This will make the data presentation easier to follow. Also, in the table header please put mutation found/not found instead of susceptible/resistance.

The authors use age groups 18-37, 38-57, 58-77, >=78. The WHO standard forms use other age groups (15-24, 25-44, 45-64, >=65. The authors may consider reanalysing their data according to these groups to better check for age-resistance correlations.

Line 30 contains a typo error. Please put 5.9% instead of 9.5%

Reviewer #2: Surveillance for drug resistance TB is very important however the manuscript has many flaws that I have highlighted within the submitted manuscript. I also think the rationale for the work done is not well articulated and moreover the amount of work done is minimal. There will be much benefit if this method has been compared with at least one traditional method for DR determination. This will allow for comparison in terms of simplicity, sensitivity and specificity as well as cost.

Other comments

1. The manuscript needs a through language review

2. Lines 31-32: what is the reference data

3. Lines 44-49: very long sentence, loosing meaning

4. Lines 54-56: A major reason is that these are the backbone of TB treatment

5. Line 67: The authors indicated in the background that they find the methods, they used to assess DR as simple and straight forward. I do not agree that detection of mutations using melting curves is simpler in the Africa context

6. The authors should be mindful that the biomarkers (315 for KatG and -8 and -15 for inhA ) that they only used to analyse for INH detects less than 90% of isoniazid resistance in Uganda. This was not discussed as a limitation of the study.

PLoS One. 2018; 13(5): e0198091.Published online 2018 May 30. doi: 10.1371/journal.pone.0198091

7. The referenced article for the primers as indicated in the methodology section does not correspond to the cited article (Poudel et al. 2012). The cited article analyzed for DR conferring mutations among INH and RIF resistance phenotypes. There was no RT-PCR analysis nor the study designed primers for DR analysis.

8. I find setting P value less than 0.005 as indication of significance of difference in measured variable too high. I suggest the authors need to review the statistical analysis again. I think the burden of DR in TB is very crucial, that one needs to be critical in following leads

9. The authors need to reduce the amount of results repeated within the discussion. Also the discussion has too many speculations

6. PLOS authors have the option to publish the peer review history of their article (what does this mean?). If published, this will include your full peer review and any attached files.

Reviewer #1: No

Reviewer #2: No

---

## [Author Response · Author response to Decision Letter 0]

22 Jun 2021

I have attached a rebuttal letter to answer all the queries raised - please see end of PDF

---

## [Decision Letter · Decision Letter 1]

9 Aug 2021

PONE-D-21-11660R1

Rifampicin and Isoniazid drug resistance among patients diagnosed withpulmonary tuberculosis in southwestern Uganda

PLOS ONE

Dear Dr. Bazira,

Thank you for submitting your manuscript to PLOS ONE. After careful consideration, we feel that it has merit but does not fully meet PLOS ONE’s publication criteria as it currently stands. Therefore, we invite you to submit a revised version of the manuscript that addresses the points raised during the review process.

We look forward to receiving your revised manuscript.

Kind regards,

Christophe Sola, Pharm.D. Ph.D.

Academic Editor

PLOS ONE

Journal Requirements:

Reviewers' comments:

Reviewer's Responses to Questions

**Comments to the Author**

1. If the authors have adequately addressed your comments raised in a previous round of review and you feel that this manuscript is now acceptable for publication, you may indicate that here to bypass the “Comments to the Author” section, enter your conflict of interest statement in the “Confidential to Editor” section, and submit your "Accept" recommendation.

Reviewer #1: All comments have been addressed

Reviewer #2: (No Response)

2. Is the manuscript technically sound, and do the data support the conclusions?

Reviewer #1: Partly

Reviewer #2: Yes

3. Has the statistical analysis been performed appropriately and rigorously? 

Reviewer #1: I Don't Know

Reviewer #2: Yes

4. Have the authors made all data underlying the findings in their manuscript fully available?

Reviewer #1: Yes

Reviewer #2: Yes

5. Is the manuscript presented in an intelligible fashion and written in standard English?

Reviewer #1: Yes

Reviewer #2: Yes

6. Review Comments to the Author

Reviewer #1: The authors may consider the following points:

Line 22-27: Consider rephrasing, like for example:

A total of 283 sputum samples (266 from newly diagnosed and 17 from previously treated patients), collected between May 2018 and April 2019 at four different TB diagnostic centres, were assessed for RIF and INH resistance using high-resolution melt curve analysis. The overall prevalence of monoresistance to INH and RIF was 8.5% and 11% respectively, while prevalence of MDR-TB was 6.7%.

29-31: Consider rephrasing, like for example:

Furthermore, among the newly diagnosed patients, prevalence of resistance to INH, RIF and MDR-TB was 8.6%, 10.2% and 6.4% respectively; while among the previously treated cases, these prevalence rates were 5.9%, 23.5% and 11.8%.

Line 29: “8.6% had resistance to INH” Please specify: Is this monoresistance or any resistance?

Line 134: Remove “suspects”. These patients are confirmed for TB as stated in the methods section, by smear microscopy and/or Xpert.

Lines 138-147: Please consider rephrasing, it is difficult to follow. Use either % or “per cent”. For example:

Monoresistance to rifampicin and isoniazid was found in 11% (95% CI: 0.077-0.150; p, 0.087) and 8.5% (95% CI: 0.056-0.123; p, 0.692) of all the patients, respectively. Resistance to RIF and INH among newly diagnosed patients was 10.2% and 8.6 %, while among previously treated patients, resistance to RIF and INH was 23.5% and 5.9 % respectively. Furthermore, 4.9% of the samples from newly diagnosed with INH monoresistance, were found to have mutations in the InhA region while 8.6% had mutations in the katG region, a condition that can lead to phenotypic isoniazid drug resistance (Palomino 2009). The overall resistance to both drugs (MDR-TB) was 6.7% (95% CI: 0.026-0.075; p, 0.982) among all patients (6.4 % among newly diagnosed and 11.8% among previously treated patients). Of the cases with MDR-TB, 6.4% had mutations in both the rpoB and KatG regions, while 3.9% had mutations in both the rpoB and InhA sites

Lines 216-217: This sentence is not correct: “settings with an MDR-TB prevalence of more than 3% especially among previously treated patients are considered as having a high MDR-TB burden”

Reviewer #2: line 41-43: the definition for acquired and primary resistance is wrong. Primary resistance a newly diagnosed case that was infected with a resistance strain and the reverse is true for acquired.

7. PLOS authors have the option to publish the peer review history of their article (what does this mean?). If published, this will include your full peer review and any attached files.

Reviewer #1: No

Reviewer #2: No

---

## [Author Response · Author response to Decision Letter 1]

9 Sep 2021

A revised version of the manuscript that addresses the points raised during the review process has been uploaded

---

## [Editor Report · Decision Letter 2]

18 Oct 2021

Rifampicin and Isoniazid drug resistance among patients diagnosed withpulmonary tuberculosis in southwestern Uganda

PONE-D-21-11660R2

Dear Dr. Bazira

We’re pleased to inform you that your manuscript has been judged scientifically suitable for publication and will be formally accepted for publication once it meets all outstanding technical requirements.

Kind regards,

Christophe Sola, Pharm.D. Ph.D.

Academic Editor

PLOS ONE

Additional Editor Comments (optional):

lane 154-155 : "This relation, however, was not statistically significant ", please add to this sentence : "given the very low sample size of the previously treated patients sample"
---

## [Editor Report · Acceptance letter]

21 Oct 2021

PONE-D-21-11660R2 

Rifampicin and Isoniazid drug resistance among patients diagnosed with pulmonary tuberculosis in southwestern Uganda 

Dear Dr. Bazira:

I'm pleased to inform you that your manuscript has been deemed suitable for publication in PLOS ONE. Congratulations! Your manuscript is now with our production department. 

Kind regards, 

on behalf of

Pr. Christophe Sola 

Academic Editor

PLOS ONE